# In Vivo Applications of Molecularly Imprinted Polymers for Drug Delivery: A Pharmaceutical Perspective

**DOI:** 10.3390/ijms232214071

**Published:** 2022-11-15

**Authors:** Alexandra-Iulia Bărăian, Bogdan-Cezar Iacob, Andreea Elena Bodoki, Ede Bodoki

**Affiliations:** 1Department of Analytical Chemistry, “Iuliu Hațieganu” University of Medicine and Pharmacy, 400349 Cluj-Napoca, Romania; 2Department of General and Inorganic Chemistry, “Iuliu Hațieganu” University of Medicine and Pharmacy, 400010 Cluj-Napoca, Romania

**Keywords:** molecularly imprinted polymers, drug delivery systems, in vivo research, drug reservoirs, targeted delivery

## Abstract

Molecularly imprinted polymers (MIPs) have been proven to be a promising candidate for drug delivery systems (DDS) due to their ability to provide a sustained and controlled drug release, making them useful for treating a wide range of medical conditions. MIP-based DDS offer many advantages, including the administration of a smaller drug doses, due to the higher drug payload or targeted delivery, resulting in fewer side effects, as well as the possibility of attaining high concentrations of the drug in the targeted tissues. Whether designed as drug reservoirs or targeted DDS, MIPs are of great value to drug delivery as conventional drug formulations can be redesigned as DDS to overcome the active pharmaceutical ingredient’s (APIs) poor bioavailability, toxic effects, or other shortcomings that previously made them less efficient or unsuitable for therapy. Therefore, MIP design could be a promising alternative to the challenging research and development of new lead compounds. Research on MIPs is primarily conducted from a material science perspective, which often overlooks some of their key pharmaceutical requirements. In this review, we emphasize the specific features that make MIPs suitable for clinical use, from both a material science and a biopharmaceutical perspective.

## 1. Introduction

The development of new materials and technologies is indispensable for research in the field of nanotechnology. Molecularly imprinted polymers (MIPs) have become the subject of pharmaceutical research due to their favorable properties and wide range of applications in biomedicine [1]. The versatility of MIPs makes them useful as diagnostic or imaging tools, drug delivery systems (DDS) or biosensing elements [2].

MIPs, also known as artificial antibodies, have drawn the attention of the research community due to their numerous advantages, such as selective binding of target molecules, increased drug loading capacities, biocompatibility and biodegradability, reduced toxicity, low-cost synthesis, and various routes of administration. In the fabrication process of MIPs, a certain template molecule interacts with one or multiple functional monomers via covalent or non-covalent bonds, followed by polymerization in the presence of a cross-linking agent. Upon template removal, specific binding cavities are left behind that are complementary to the molecule in terms of shape, size, and functionality [3].

MIPs have become popular for drug delivery due to their versatility in terms of administration routes, high loading capacity, and (stereo)selectivity for the desired target. Their superior drug loading, compared to conventional non-imprinted polymeric systems, leads to lower dosages and, thus, to a reduced susceptibility to adverse reactions and better safety profiles [4]. Despite their low immunogenic characteristics, MIPs biocompatibility is still a controversial subject. It is generally accepted that MIPs offer excellent biocompatibility, however, their long-term effects on living organisms have not been thoroughly investigated yet [5].

From a drug delivery perspective, MIPs fall into two categories: drug reservoirs and targeted DDS (Figure 1). Targeted DDS are administered intravenously and are intended to cross certain biological barriers in order to reach a desired targeted tissue, organ or epitope. In contrast, drug reservoirs were not designed to bypass biological barriers and remain confined at or within the vicinity of the administration site, wherein the drug could diffuse in the surrounding tissue, or even systemically distributed via the circulatory system.

Drug reservoirs provide a sustained or controlled drug release, high drug loading capacity and low frequency of administration, whilst also lacking size restrictions, acute and long-term toxicity. On the other hand, targeted injectable DDS slightly differ from drug reservoirs and are characterized by built-in or external guiding towards a targeted tissue/organ, stimuli-responsive drug release (by photodynamic activation, local heating, pH change, biomarker fluctuations, etc.), size restrictions are imposed to avoid biological barriers, vascular occlusion, and premature clearance. Targeted DDS do not present immunogenicity, acute or long-term toxic effects, and can be designed to be biodegradable. Moreover, they are useful for medical diagnosis, but can only be administered intravenously.

Biodegradable polymers have recently come into the spotlight in the drug delivery field as the process of biodegradation not only improves the release kinetics, but also offers a superior safety profile. Given that the polymeric network is broken down and eroded, they are less susceptible to accumulating in organs, hence being more biocompatible [6]. Recent studies have shown that, in vivo applications of MIPs, the formation of protein corona (PC) should also be taken into consideration as this possibly alters their bioavailability and efficacy [7].

The primary aim of this article is to critically discuss and compare MIPs designed for drug delivery, focusing only on the materials that have reached the in vivo stage. The majority of research conducted in the field of MIPs is conducted from a material science perspective, in which some of their pharmaceutically relevant aspects are often omitted. This validates and justifies the necessity of a review that integrates both the material science and biopharmaceutical perspectives, with an emphasis on the critical features that decide upon MIPs potential clinical use. As MIP design has been marked by the lack of an interdisciplinary approach, we aimed to review the most recent publications in the field from a larger perspective, bringing together engineered functional (nano)materials and biomedical sciences.

## 2. Background and History of MIPs

Molecularly imprinting technique (MIT) was first reported in 1972 by Wulff and Sarhan [8], and has subsequently been adopted by many research groups. The preparation of MIPs has the benefit of being simple, fast, economical, and robust, however, some challenges remain. One of the most important aspects to be considered, if opting for MIT, is the proper selection of the template, solvent and functional monomer, all guided by the nature of application [9]. Monomer selection depends on the physical-chemical properties of the template, so that its functional groups can interact with the monomer to form stable supramolecular complexes, involving non-covalent, coordinative, or covalent bonding. The stability of the prepolymerization complex is promoted by either acid-base functional complementarity, hydrogen bond donor-acceptor, or other types of weak interactions between the participating molecules, i.e., template and functional monomers. Usually, a 1:4 template:monomer ratio provides an acceptable stability for the complex [10]. Figure 2 illustrates the simplified MIP synthesis.

Polymerization also requires cross-linkers to ensure the structural stability of the polymeric scaffold and to increase its porosity. The type of cross-linker is related to the preservation of the binding cavity’s shape, as it directly influences the physico-chemical and mechanical properties of the polymers. The most common cross-linker is ethylene glycol dimethacrylate (EGDMA), due to the increased mechanical stability provided. Other possible cross-linkers are divinylbenzene (DVB) or trimethylolpropane trimethacrylate (TRIM). Generally, the resulting MIPs are characterized by high structural robustness in a wide range of pH, solvents and temperature [10]. Furthermore, the nature and concentration of a cross-linker is closely connected to other characteristics. A low concentration of cross-linker leads to poor stability and the faster release of the template, while hydrophilic agents increase biocompatibility [5].

The imprinting process is generally similar when designing MIPs for either analytical applications or drug delivery. However, each application requires a specific synthetic approach. MIPs were initially developed for analytical purposes; therefore, most of the later designed polymers intended for drug delivery were made using the same acrylic monomers, via non-covalent imprinting. The main adjustment when changing the field from analytical purposes to drug delivery is a decrease in the cross-linker ratio. For analytical applications, such as molecular recognition, sensing, or separations, MIPs should be highly cross-linked to ensure a highly specific rebinding of the template molecule through their rigid cavities. For drug delivery, however, a lower degree of cross-linking is desirable in order to provide tunable release kinetics. While the imprinting factor is crucial for MIPs employed in analysis, their drug loading capacity becomes more relevant in the field of drug delivery. These features are interconnected, as a higher binding affinity might imply a higher drug loading capacity and extended drug release. Selected monomers, cross-linkers, and porogens for drug delivery should be biocompatible and biodegradable in order to reduce the possible negative impact on the viability of healthy cells [4,11].

The solvent also plays a crucial role in the process, as it directly influences the strength of the template monomer interaction. Solvents that have low dielectric constants are usually preferred (chloroform or toluene), as they tend to stabilize the electrostatic interactions and hydrogen bonds. Solvents with higher dielectric constants (acetonitrile) may also be used, although the resulting MIPs will have a lower affinity to rebind the template [10]. Whilst they pose great concerns regarding environmental toxicity, the conventional solvents used for MIP synthesis are acetonitrile, methanol, chloroform, tetrahydrofuran, toluene, dichloroethane and *N*,*N*-dimethylformamide [12].

The solvent’s nature is a general quality criterion frequently omitted from studies, which becomes highly relevant when studying MIPs in vivo. Most MIPs intended for drug delivery applications are fabricated with the use of aprotic and low polarity organic solvents for the preservation of hydrogen bonds. Due to the possibility of additional or unexpected toxicity issues, residual organic solvents are unsuitable for drug delivery applications. In addition to their potential for inducing cellular damage, porogenic solvents also influence the polymer’s morphology, and could negatively affect their release kinetics. Hence, an aqueous media would be ideal, and is sometimes required, for conducting MIP synthesis [11].

Moving from solution-based synthesis to solid-phase synthesis (SPS) represented another important milestone in the development of MIP-based DDS. In SPS, the template molecule is immobilized on a solid phase and polymerization occurs around it, generating standardized imprinted nanoparticles. The main benefit of using SPS is the low polydispersity index of the resulting formulation and a more homogenous distribution of the binding sites, which occurs as a consequence of the template’s orientation on the solid phase. Moreover, the covalent surface immobilization of the template enables the removal of the low-binding polymers prior to the elution of the imprinted particles, while offering a more homogenous binding affinity to the latter fraction, as well as the facile reusability of the surface bound template. Therefore, SPS has been successfully adopted for the fabrication of nanoMIPs with reproducible features and enhanced release kinetics [5].

Although an empirical approach has, until recently, been used for the fabrication of MIPs, new methods for a more rational design have emerged. These include computational simulation-aided methods and rational algorithms. Computational methods combine quantum and statistical mechanics to simulate static or dynamic molecular interactions. Their advantages include cost-effectiveness, prediction of template-monomer conformational interactions, simulation of the appropriate composition and molar ratio of the pre-polymerization mixture. The methods used for theoretical simulations in MIP design include molecular mechanics (MM) [13], molecular dynamics (MD) [14], and quantum mechanics (QM) [15]. QM provides the highest accuracy for choosing the initial direction of interacting molecules, while MM is the most time- and cost-effective in terms of reagent and solvent use. The complexity of calculations expands exponentially with QM; thus, the most widely used method for complex mixtures is MD. Through MD, one can optimize the molar ratio between the template, monomer, and cross-linker. Through computational methods, the binding energy between the template and functional monomer is simulated and calculated. High binding energy means high selectivity and binding characteristics of designed MIPs. Moreover, these methods make it possible to identify and study the mechanisms underlying MIP formation on a molecular basis. Molecular modelling has been widely applied for the optimization of various materials as it can lower production costs, analysis and synthesis time, as well as the consumption of organic solvents or other polluting reagents. The rational design of MIPs is also based on the nature of the template. If the template is a macromolecule, simulation with QM is not feasible due to the high computational time and risk for errors. With MM and MD, the operation is faster and more accurate for complex mixtures or imprinted templates [16].

When considering the nature of the imprinted template, the process for small molecules is somewhat robust and reliable. For macromolecules, however, due to their conformational flexibility, suboptimal selectivity and affinity of the resulting imprinted sites are often reported. Moreover, template-assisted imprinting of exotic biomacromolecules may be excessively costly, whereas, removing from or rebinding by the imprinted polymer large, bulky macromolecules is also accompanied by severe limitations. To address these shortcomings, alternative methods, such as epitope and surface imprinting, have been proposed and developed over the last two decades in MIP-based purification and sensing [17,18,19]. Evidently, sustained efforts are also being made towards rational approaches in selecting the most suitable and representative short sequence of oligopeptide (epitope) as a template equivalent with the ability to efficiently and reproducibly generate selective binding sites with homogenous affinity towards the native protein target [20].

The first designed MIPs for drug delivery applications were based on imprinting theophylline with methacrylic acid (MAA) as the functional monomer and EGDMA as the cross-linker [21]. In addition to the good recognition properties for theophylline, compared to its structural analogue caffeine, these MIPs could prolong the release of theophylline [5]. The first MIP to be administered and studied in vivo conditions was designed by Hoshino et al. in 2010. Their research group developed a MIP-based platform for the recognition, neutralization, and removal of peptides [22]. Additionally, the first MIP that has been specifically used for drug delivery and studied in vivo on mice was developed by Wu et al. [23]. Their work, published in 2015, focused on the therapy of the *Helicobacter*
*pylori* infection treated by amoxicillin-loaded MIPs [23]. In recent years, an increasing trend in the administration and study of MIPs in living organisms is observed. However, the issue of their biocompatibility is still a subject of controversy, and their long-term safety is yet to be evaluated and discussed.

## 3. In Vivo Applications of MIP-Based DDS

MIPs are promising DDSs due to their ability to provide sustained or controlled release and targeted delivery of a selected drug, related to their high binding affinity to the template. To date, various MIP-DDS have been developed for oral, intravenous, ocular, or transdermal administration. They were effective in various pathologies, such as: cancer, arrhythmias, avitaminosis, cardiovascular and brain disease, inflammatory disorders and addiction therapy [1]. Using MIPs as controlled DDS has a number of advantages, such as the administration of lower drug doses due to the high drug loading or targeted delivery, thus reducing any useless spread in the body. This leads to fewer side effects, along with the possibility of attaining high concentrations of the drug in the targeted tissues [24]. Their advantages are illustrated in Figure 3.

Oncology is one of the major fields in which functional polymers can bring significant benefits. The use of MIPs plays an important role in the efficacy of chemotherapy as it can assure a targeted and controlled delivery of antitumor agents. By incorporating chemotherapy agents into MIPs, their release kinetics is significantly improved [11].

There are three possible mechanisms of drug release from polymeric matrices: diffusion, erosion, and desorption of a surface-adsorbed drug. When the drug is homogenously dispersed in the matrix, release occurs due to diffusion, while erosion takes place when using a (bio)degradable matrix. An initial burst release of the drug occurs through a fast initial desorption process from MIP’s surface [25]. Imprinted drug reservoirs can provide a sustained zero-order release, for a long period of time, while protecting the active ingredient from enzymatic, hydrolytic or photo-degradation, thus increasing its bioavailability [4].

Most conventional DDS show major limitations that negatively affect the activity of chemotherapeutic agents for various reasons, such as a decreased loading capacity or burst release of the embedded molecule [26]. MIPs, however, can be designed to release their payload as feed-back to external stimuli [5]. A smart MIP-based DDS can be fabricated using chemicals with magnetic properties, materials sensitive to pH, temperature, biological macromolecules or even UV light. Combining MIT and stimuli-responsiveness can improve the absorption and release profile through changes in the environment [1]. Variations in the pH is the most commonly used strategy for the development of MIP-DDS for cancer therapy. Acidic pH values in the tumor environment can be exploited to break the bonds in the DDS, finally leading to an increased drug release [11]. As highlighted by Konstantin et. al. in their most recent review, MIPs have become promising platforms for both the in vitro and in vivo administration of therapeutical agents. They exceed the limitations of “natural antibodies” in terms of synthesis costs, immunogenicity and stability, thus being preferred for potential clinical applications [5].

From a fit-for-purpose perspective, MIPs employed in drug delivery applications can be broadly divided into two categories: MIP-based reservoirs and imprinted injectable nano-systems. Their critical characteristics are summarized and compared in Table 1. The first type of DDS is represented by the conventional bulk solid MIPs or hydrogels, or by the micron-sized imprinted particles that are not carried out by the vascular system. Whether external to the body or implanted, this type of DDS is generally intended to be administered once or sporadically in a specific region of the body, for a long-term local or systemic activity. High drug loading capacities are desirable, so that lower amounts of vehicles need to be implanted. Stimuli-responsiveness is another optional feature that allows on-demand release of the payload, dependent either on the local/endogenous stimuli such as pH, redox gradient, temperature, biomarker concentration, or on external/exogenous stimuli, such as light, ultrasound, electric or magnetic field. Employing the imprinting technology in the fabrication of drug delivery reservoirs has a range of potential benefits, including high drug loading efficacy, better protection for loaded drugs, economical and straightforward synthesis.

The imprinted nano-systems are designed to be intravenously administered, while being equipped with active targeting mechanism(s) based on built-in (natural receptor, epitope imprints, aptamers) or external (magnetic field) guiding towards a targeted tissue or organ. In addition to the specific delivery at the action site, payload delivery can also be controlled by a feedback release based on internal or external stimuli. Unfortunately, the injectable administration of DDS involves several immunological risks with serious clinical implications, therefore immunotoxicity must be investigated.

The particularities of drug release mechanisms are related to the type of DDS. In the case of drug reservoirs, release can occur by diffusion, mechanical erosion, competitive displacement, or stimuli-triggering. In contrast, drug release from nanoMIPs can only take place by diffusion or as feed-back to stimuli.

In all cases of internally administered polymers, different concerns over their safety need to be addressed, especially after long-term administration. These issues include biodegradability, natural clearance, circulating instability and the lack of acute and long-term toxicity.

A few studies regarding the safety and efficacy of MIPs in vivo are to be discussed below from the fit-for-purpose perspective, and their most important characteristics are summarized in Table 2.

### 3.1. Drug Reservoirs

When addressing the design of drug reservoirs, the range of templates of interest to be imprinted is very broad. Some of the typical templates include small organic molecules such as pesticides, active pharmaceutical ingredients (APIs), aminoacids, peptides, and glucides. However, novel strategies recently emerged for imprinting larger compounds, such as proteins, cells, bacteria, and viruses [12]. Insulin has been successfully included in a biomimetic MIP-DDS for oral administration, as an alternative to the traditional subcutaneous formulation for diabetes [27]. Due to its peptide structure, insulin can easily benefit from being embedded in a MIP, as it is readily degradable in the gastro-intestinal tract. N-hydroxyethyl acrylamide (HEAA)-based MIPs exhibited an enhanced hypoglycemic effect in vivo, quantified by 60% reduction in blood glucose levels in the first 2–3 h. Surprisingly, no secondary hypoglycemic adverse effects were noticed, subsequently suggesting the favorable safety profile of this biomimetic platform [27].

Enantioselective MIPs are also potential strategies for designing systems with a controlled delivery. More than a decade ago, Suedee et al. developed and evaluated the performance of a transdermal system for the enantioselective-controlled delivery of S-propranolol, which consisted of four components: the backing layer, a chitosan gel as the reservoir, a MIP-based membrane, and the release liner. Chitosan was selected as the gel vehicle due to its favorable characteristics, such as high selectivity for the S-enantiomer release, low toxicity, and reduced immunogenicity. An intriguing aspect regarding this study was the preparation of MIP-based membrane by reactive pore filling of a bacterial cellulose membrane. The enantioselective-controlled delivery of S-propranolol from the racemic propranolol-loaded gel formulation was based on the membrane’s high selectivity towards the eutomer. In vitro release tests revealed that an increase in the pH of the receiving solution enhances the enantioselective release of S-propranolol. Explanation lies in the fact that increasing the pH leads to a higher degree of ionization of the monomer, causing the stronger affinity towards the S-enantiomer at the recognition sites, promoting its transfer through the stereoselective membrane. The following skin permeation studies surprisingly revealed that cellulose alone provides a low to moderate natural-occurring selectivity for the enantioselective release of propranolol. Finally, the transdermal patch has proven its in vivo effectiveness as a drug reservoir due to the ability to selectively regulate the release of S-propranolol, with a limited transport of the distomer [28].

Oral DDS formulations can be improved by designing systems with superior floating properties for the improved bioavailability (BA) of a desired drug. The floatation approach implies that the drug reservoir must exhibit a lower density compared to gastric fluids, which leads to its increased gastric residence time following oral administration. This can be achieved by using a liquid crystal (LC), 4-methylphenyl dicyclohexyl ethylene (MPDE) as functional monomer [29].

LC-MIPs can imprint and recognize templates at a very low level of crosslinking. Compared to conventional MIPs, they have a higher number of available binding sites, which can overcome the challenge of the low employment of imprinting sites. Due to a decrease in the cross-linking level, the mass transfer of templates can be significantly improved, while superior release profiles are achieved. However, the imprinting effect of LC-MIPs is usually weaker due to the low degree of cross-linking [30]. In one study, MPDE and polyhedral oligomeric silsesquioxanes (POSS) were used to fabricate a floating MIP-DDS for the oral administration of capecitabine (CAP). Throughout in vitro and in vivo studies, the therapeutic system exhibited a sustained release and superior BA of CAP, consequences to its increased gastric floatation effect in aqueous media [29].

Nanostructured DDS based on carbon nanotubes (CNTs) have also taken the spotlight recently. CNTs consist of hexagonal arrangements of carbon atoms, resulting in cylindrical nanostructures. Based on the layer of graphene sheets, CNTs fall into two categories: single-walled (SWCNTs), with the outer diameter of 0.4–2 nm, and multi-walled CNTs (MWCNTs), with the outer diameter ranging between 10 and 100 nm.

CNTs are defined by special distinctive electrical, mechanical and optical characteristics. They are able to bind biological molecules due to their high surface area. Their inner hollow structure is usually used to load certain drugs, while the outer surface can be chemically or physically modified by adsorption, electrostatic interactions or covalent bonds to increase their hydrophilicity [31]. One concern regarding the use of CNTs in living organisms is their significant toxicity observed in vivo conditions, due to their hydrophobic surface and limited solubility in aqueous media. CNTs are considered responsible for many harmful effects, such as free radicals and reactive oxygen species (ROS) formation, apoptosis, and inflammatory effects. To overcome this limitation, CNTs should be functionalized (such as 9-vinylanthracene, 9-VA) to enhance their clearance and thus lower their toxicity [31,32].

CNTs advantages include: increased surface area, lightness, thermal stability, lack of swelling and high stability in acidic media. They can be associated with LC for oral floating drug reservoirs, with metal-organic framework (MOF) gels for facile dispersion, or even fabricated in binary green porogen systems to avoid the health hazards of conventional solvents [32,33,34]. Zhang et al. designed a floating device based on MWCNTs, coated with LC-MIPs for the oral delivery of levofloxacin. To bypass the potential safety threats, the nanotubes were functionalized with 9-VA to enhance their clearance. MIP synthesis was based on the “grafting to” approach, relying on strong non-covalent bonds which occur at the surface of MIPs due to the π-π interactions of vinyl groups. In vivo pharmacokinetic studies revealed an increased floating time of more than 24 h, along with an impressive relative BA of 578.9%, advising in favor of their potential future applicability in the field of gastro-retentive DDS [20].

MOFs selected for designing MIP-based drug reservoirs offered superior release profiles. MOFs, also known as coordination polymers, are metallic organic frameworks in which metal ions (Cr^3+^, Fe^3+^) are bound together by organic ligands (e.g., polycarboxylic acids). The resulting nanostructured materials have a few unique properties, such as high porosity, conductivity, catalytic activity, alongside the possibility to be molecularly imprinted. These distinctive characteristics have brought MOGs significant attention from the research community, as they find applications in various fields [34]. In one study, Fe (III)-trimesic acid was selected as the dispersant for MIP-based DDS, doped with MWCNTs, aiming at the controlled release of aminoglutethimide (AG) in breast carcinoma therapy. Preliminary tests revealed a significant increase in the NPs’ specific area and pore volume, due to the contribution effect of MWCNTs and MOGs, offering proper porosity and stability to the gels. Interestingly, it was observed that MIP-based MOGs lost their recognition ability after metal ions removal. In vitro and in vivo studies confirmed that MIPs affinity towards the template can be significantly improved by including both MWCNTs and MOGs in the formulation, which are responsible for the enhanced adsorption, controlled release and superior BA of the desired drug [34].

Room temperature ionic liquids (RTILs) are a class of solvents with impressive features. They can act as porogenic agents in the polymerization process and lower MIPs shrinkage in conventional porogens, due to their low vapor pressure. RTILs and deep eutectic solvents (DESs) can be associated as dispersing media, aiming to stabilize and prevent CNTs reaggregation and increasing template’s solubility. Replacing conventional porogens with “green solvents” reduces the dangers they exhibit on humans and environment [12]. This approach has been selected for the fabrication of a controlled-release powder with fenbufen (FB) imprints made of single-walled carbon nanotubes (SWCNTs)-doping MIP nanocomposite-based binary green porogen system. The binary system consisted of RTIL 1-butyl-3-methylimidazolium tetrafluoroborate ([BMIM]BF4), which acts as a dispersing media for CNTs, and DES choline chloride/ethylene glycol (ChCl/EG). Choosing FB as the template was based on the premise that its inclusion in a DDS could reduce its harmful gastro-intestinal side effects. The complex drug reservoir proved to be highly effective in vivo, offering a relative bioavailability of 143.3% [33].

Another approach for the design of MIPs is to use pH and temperature changes as external stimuli for improved drug delivery. Hydrogel-based MIPs (hydroMIPs) are materials that integrate MIPs and a stimuli-responsive hydrogel, resulting in polymers that can reversibly shrink or swell in response to environmental changes (pH, temperature, enzymes). For example, Wang et al. developed an original thermoresponsive hydroMIP-DDS for the delivery of gatifloxacin (GTX). Herein, frontal polymerization (FP) technique was tested for MIP synthesis, an alternative that allows the conversion of monomer into polymer by using the heat released from the polymerization reaction to create a self-sustaining front that propagates throughout an unstirred monomeric mixture. FP has many advantages compared to other techniques: shorter reaction time, low-energy consumption, homogeneity enhancement of polymer chains. Moreover, hydrogels prepared by FP exhibit higher swelling rates and ratios compared to those prepared by bulk polymerization. The most common thermo-responsive polymer is poly(N-isopropylacrylamide) (pNIPAm) because its low cloud point temperature (CPT) of 32 °C. PNIPAm undergoes a reversible phase transition in water, changing from soluble chains below CPT to hydrophobic aggregates above it. Hydrogel imprinting brings significant improvements over conventional hydrogels, in terms of binding affinity, loading capacity, and selectivity towards various templates. The pH-responsiveness of the gel was shown by a significantly lower drug release in acidic medium, and faster release at higher temperatures (43 °C), suggesting that the imprinting effect may also be temperature dependent [35].

Solvent-responsiveness is another technique used for obtaining controlled-release drug reservoirs. For instance, LC is capable of solvent-responsive deformation, controlled entirely by the nature of the solvent environment (miscibility, polarity, and hydrogen bonding). A solvent-responsive floating LC-MIP was developed recently, aiming at the gastroretentive controlled release of S-amlodipine (S-AML). As expected, in vivo studies showed that floating LC-MIPs exhibit a prolonged gastric residence time, over 60 min, and high bioavailability. The floating behavior of LC-MIPs in aqueous medium was attributed of their solvent-responsive deformation [36].

### 3.2. Targeted DDS

MIPs are particularly useful in targeted tumor therapy. In addition to loading cytostatic drugs, MIPs can be used for imprinting specific epitopes for the recognition of cancer cells that overexpress specific biomarkers. The resulting dual-imprinted DDS have the ability to specifically recognize and target cancer cells, along with providing a sustained drug delivery at the targeted tissue/organ. Moreover, fluorescent materials or paramagnetic metals can be associated, leading to highly efficient theranostic platforms that combine imaging and targeted drug delivery for safer and more efficient therapeutics [5].

The double-imprinting approach was adopted by Qin et al. for the development of fluorescent MIPs (FMIPs), intended for both targeted recognition and drug delivery. FMIPs were designed by imprinting two different templates: doxorubicin (DOX) and the N-helix terminal epitope of P32 protein, a membrane receptor that is overexpressed on the surface of 4T1 breast cancer cells. DOX-loaded cavities aimed to provide its sustained release, thus reducing potential harmful effects on healthy cells, while the epitope-imprinted sites were meant to specifically recognize, target, and kill the tumor cells which overexpress P32. The core-shell structure design of NPs consisted of an outer imprinted polymer layer prepared via the surface double-imprinting technique, alongside a nucleus-embedded silica NPs. In addition to the numerous advantages of these NPs bringing to the formulation, such as low toxicity, mechanical stability, high solubility, and biocompatibility, it also offers excellent fluorescent properties which subsequently result in the high efficiency of MIPs for targeted fluorescence imaging. In vitro experiments highlighted the major influence pH media had on tumor-targeted drug release, outlined by a 2.7 times greater release of DOX in the typical tumor acidic microenvironment (pH = 6) versus the neutral media (pH = 7.4). Moreover, in vitro cytotoxicity assay emphasized the targeted system’s low toxicity and biocompatible characteristics attributed to the silica-core NPs. The complex platform exhibited superior in vivo anticancer effects described by an 0.8 times higher tumor reduction and subsequently revealed a major resemblance between the intravenous and intratumor routes regarding its antitumor efficacy [37].

Paramagnetic metals can also be added to the core-shell NPs for magnetic resonance imaging, gadolinium (Gd) being the most frequently used. Photosensitizers can be additionally loaded for an enhanced therapeutic effect. One paper follows the design and evaluation of a complex MIP-DDS intended for chemo-/photodynamic synergistic targeted cancer therapy. The dual-imprinting technique was selected to obtain MIPs that simultaneously deliver DOX and target an overexpressed biomarker, CD59 protein, represented herein by its epitope. Moreover, the platform was loaded with gadolinium-doped silicon quantum dots for fluorescent/magnetic resonance imaging, along with the chlorin e6 (Ce6) photosensitizer agent. Ce6’s mechanism is based on generating reactive oxygen species when exposed to laser irradiation (655 nm). These radicals will exhibit localized cytotoxic effects and subsequently act synergistically with DOX by inducing cancer cell apoptosis. Along with the low toxic effects observed in both in vivo/vitro conditions, the DDS possess high biomarker-targeting activity, a superior therapeutic efficacy and a tolerable safety profile, assets that continue to advise in favor of their use as a promising tool in oncology [38].

The properties of MIPs can easily be adjusted by introducing additional co-monomers, such as HEAA, for faster release kinetics in neutral medium [27], or 4-vinylbenzeneboronic acid (4-VBBA), for improved imprinting effect [39]. The cooperation effect of 4-VBBA and MAA on the affinity of CAP imprinted MIPs was demonstrated by Yuan et al. [39]. The association of 4-VBBA and MAA was thought to exhibit a higher affinity for the template. Preliminary results confirmed this hypothesis by demonstrating a high imprinting factor of the imprinted polymer, which consequently suggests superior binding and recognition properties towards the template. Moreover, the improved bioavailability was attributed to the boronic acid fraction [39].

One of the most frequently adopted methods for the external steering of MIP-DDSs is through magnetic field guiding [40]. A magnetic MIP-DDS is most commonly made of an iron oxide nanoparticle core and a drug-loaded MIP shell, delivered to the tumor site with the aid of an external magnetic field. They can be also coated with different polymeric materials to improve their biocompatibility. For instance, DOX loaded magnetic MIPs coated with polydopamine were very effective in a mouse breast adenocarcinoma model. Polydopamine was selected because of its high biocompatibility, with the purpose of increasing the NPs stability against oxidation. In vivo studies proved their superior therapeutic effect, high survival rates in mice, and good biocompatibility [40].

Stimuli-responsive MIPs are a new generation of intelligent therapeutic systems that can adjust their properties in response to various external or internal stimuli [41]. Although such DDSs have been successfully validated under in vitro conditions [42,43], unfortunately this level of valued potential has not yet been demonstrated in vivo.

The surface functionalization of imprinted nanoparticles with neutral hydrophilic polymers, such as poly (ethylene glycol) (PEG), would assist such nanomaterials in evading the immune system by modifying the nature and number of proteins that are adsorbed on their surface, giving rise to the so-called PC. PC refers to a layer of proteins that covers the administered nanocarrier, formed through the interaction between circulating proteins and the delivery system. Similar strategies to manipulate PC formation and to endow an immune stealth feature to non-imprinted nanosized DDSs imply the controlled exposure to endogenous de-opsonin proteins (human serum albumin, transferrin, apolipoprotein E) prior to intravenous administration [44]. Additionally, such pretreatments would allow the extension of blood circulation time of DDSs leading to enhanced bioavailability of the API.

Biodegradable building blocks could further contribute to the much-desired biocompatibility, ensuring the lack of acute and long-term toxicity of such imprinted nanocarriers. For example, fructose was used as a biodegradable monomer/cross-linker for the controlled delivery of olanzapine to the brain, by means of magnetic iron oxide-coated silica NPs. In addition to its high biocompatibility, fructose was selected because it can be used as an energy source by the healthy braincells after degradation. The targeted DDS showed high selectivity and biocompatibility, controllable performance, and low toxicity [45]. Similarly, tannic acid was used as a monomer/cross-linker for the fabrication of a magnetic fluorescent multi core-shell structure MIPs-based NPs, aiming to offer a sustained and targeted delivery of 5-fluorouracil (5-FU) in cancer treatment. Tannic acid is a biodegradable polyphenolic compound with potential anticancer effects. Some of its many benefits include high biodegradability, low cost, high antioxidant capacity, antimutagenic and antimicrobial properties. Surprisingly, tannic acid is also a natural cross-linker due to the hydroxyl and carboxyl groups that can efficiently interact with the polymeric backbone. In vivo studies showed that both 5-FU magnetic MIPs and tannic acid determine tumor morphology change and cell death, suggesting their anticancer efficacy as targeted DDS [46].

## 4. MIPs as Theranostic Platforms

MIPs can also be employed as building blocks for advanced theranostic systems. The concept of theranostics is defined by integrating both diagnostics and targeted therapy in a single platform, which leads to enhanced therapy and simultaneous imaging. This novel approach brings significant advantages to drug delivery, such as real-time treatment monitoring, real-time surgical guidance, evaluation of disease prognosis, and drug biodistribution assessment [47,48].

Given the versatility of MIPs, regarding the nature of the imprinted template, they can also be loaded with bioimaging markers. MIPs have become valuable tools, not only for therapeutic purposes, but also for diagnosing diverse pathologies. Among these, it has been reported that *Pseudomonas aeruginosa* and *Helicobacter pylori* infections can be diagnosed and treated using fluorescent MIPs by targeting specific biomarkers of the pathogens [48]. Oncology is particularly benefiting from the theranostic approach, due to the possibility of loading both cytostatic drugs and imaging compounds. This newly emerging field brings together two vital steps in cancer management: guided drug delivery and optical imaging by targeting overexpressed glycans or surface receptors of the tumoral cells [48].

Imaging methods include any scanning techniques that provide two-dimensional images, such as fluorescence, luminescence, infra-red or Raman spectroscopy, magnetic resonance imaging (MRI), radionuclide-based imaging, computer tomography (CT), positron electron tomography (PET), electrochemical, ultrasound or X-ray technology [49]. MIP-based imaging provides highly efficient localization of tumors. Imaging agents include organic dyes, silica NPs, carbon nanodots, gold and silver NPs, radioisotopes, magnetic iron oxide NPs, rhodamine B, fluorescein isothiocyanate (FTIC), gold nanorods, ^14^C-labelled acrylamide [49].

According to imagistic experts, the ideal imaging nanoparticle for clinical applications should be biodegradable or quickly cleared from the body, with no to low toxicity, and the ability to generate a strong imaging signal. The main issues with fluorescent imaging lie in the strength of the detected signal, which is directly related to the depth of the tumoral tissue of interest, and in the interference of autofluorescence coming from endogenous molecules. Therefore, one of the necessary characteristics of MIPs used in bioimaging includes the use of fluorophores that emit in the near-infrared (NIR) range, to facilitate deep tissular scanning, avoid autofluorescence, and acquire a strong fluorescent signal. Moreover, particles below 100 nm are preferred for in vivo imagistic purposes, mainly because smaller particles give higher resolution images [48].

In terms of the targeted delivery of therapeutical agents, MIPs can be designed for active or passive targeting. Active targeting implies the imprinting of a specific sequence, for example, an epitope that would specifically target an overexpressed biomarker, where the nanoMIPs would release their payload. For passive targeting, however, MIPs would have to exploit the EPR effect to reach the targeted tissue and unload the theranostic agents [48].

The application of MIPs in bioimaging is primarily focused on fluorescence techniques, due to their inherent benefits: low instrumental and operating costs and high resolution of scanned images. For this purpose, MIPs can be fabricated either by copolymerization of fluorescent monomers or loading of fluorophores. The resulting MIPs can be used for both the in vitro and in vivo study of viable cells, tissue imaging, and the assessment of drug biodistribution [48].

Quantum dots (QDs) or N-fluoresceinacrylamide-, indocyanine green-, or Cy5 NIR dye labelling are widely used for in vivo fluorescence imaging, whereas other materials have also been proven to be highly efficient, including iron or gadolinium contrast agents for MRI, ^124^I labelling for PET imaging [48,49].

Regarding fluorescence bioimaging in oncology, tumor surgical resection can be facilitated by fluorescence-guided surgery. For this purpose, an IR camera can provide surgeons with useful images for a more accurate and selective demarcation and resection. For example, hybrid silica NPs (Cornell dots) have been successfully used for imaging sentinel lymph nodes in patients with metastases [48]. A number of the studies described above successfully present the design of theragnostic platforms based on fluorescent MIPs for active targeted drug delivery and imaging, using DOX along with P32 or CD59 epitopes as templates to target cancer cells [37,38].

## 5. Biocompatibility and Biodegradability

The most crucial and often neglected property of MIPs designed for potential clinical applications is their biocompatibility, which is directly related to their surface chemistry. Initial evaluation of MIPs consists of in vitro tests studying both their specific and non-specific toxicity on living cells. In addition to the common cell viability assays, other tests are conducted to evaluate a possible inflammatory response, cellular metabolism alterations, or changes in cell function and morphology [5]. A few studies show that MIPs do not commonly present in vitro toxicity, especially if biodegradable components are used for synthesis. However, for in vivo applications, consistent and reliable long-term toxicity tests are necessary [5].

Particle size is a crucial parameter that not only determines the drug loading, release and biodistribution, but also influences their toxicity. When particle size exceeds 100 nm, MIPs can spread through the blood flow to multiple organs. The particle diameter should ideally be in the range of 10–150 nm for effective tissular/cellular penetration, longer circulating time and efficient accumulation in the targeted tissue. In the case of smaller NPs, their high surface-volume ratio is advantageous from a drug release kinetics perspective, however, they can easily aggregate in biological media. Sizes above 200 nm are usually avoided as they have a negative effect on circulation time and could possibly lead to obstruction of capillaries. Their surface charge is also involved in their toxicity, as positive NPs seem to be more toxic than their negative or close to neutral counterparts. Furthermore, positively charged particles can lead to hemolysis and clotting; therefore, anionic NPs are generally preferred [31].

Generally, after being coated by opsonins in the systemic circulation, NPs are confined by the reticuloendothelial system to be shattered. If not destroyed, MIPs could inherently accumulate in organs over long timeframes, leading to in vivo toxicity [5]. To improve their biocompatibility, MIPs can be coated with hydrophilic macromolecules, such as PEG. As hydrophilic molecules reduce the adsorption of opsonins, the blood circulation time of MIPs increases and their toxicity is reduced considerably [5].

For MIPs to be used for therapeutic purposes, the imprinted polymeric carrier must exhibit high efficacy in drug loading, transportation, and release, as well as low toxicity. Their safety profile is determined by their biocompatibility and biodegradability. In this respect, the MIPs are considered superior to other nano-systems due to the versatility of their polymeric scaffold. Gelatin, for example, is an amphiphilic macromolecule with numerous functional groups (–NH_2_, –COOH, –OH); as it has low toxicity and immunogenicity, while also being biodegradable, gelatin becomes an ideal monomer for the development of biocompatible MIPs. For example, Tang et al. successfully developed biocompatible MIPs with gelatin as functional monomer for the recognition of testosterone. Gelatin exhibited high selectivity towards the template, without any additional nonspecific toxic effects on healthy cells. Although the polymer’s purpose was the recognition and efficient sequestration of testosterone in prostate cancer cells, its aim can be easily shifted to drug delivery due to the molecularly imprinted gelatin’s high binding selectivity and biocompatibility [50].

The testing and use of biodegradable polymers has recently intensified due to the increasing awareness regarding the necessity of superior safety profiles and higher biocompatibility of these drug delivery platforms. These polymers are cleaved by hydrolysis into non-toxic metabolites that can be easily cleared from the human body. Moreover, their release rate can be regulated to obtain a controlled degradation by adjusting cross-linking density [46].

Therefore, biodegradable cross-linked materials are extremely promising for drug delivery applications in vivo. Other advantageous properties include: acceptable stability of 3D networks, elasticity, flexibility, and superior biocompatibility due to controllable drug release [45]. The amount of cross-linking agent directly influences DDS properties. High amounts might lead to significant toxicity, while low quantities lead to the fast degradation and burst release of the desired drug [45]. Other biodegradable/biocompatible materials that are extensively used for molecular imprinting include PEG, lactic acid and poly-lactic glycolic-acid [45]. Fully degradable PLGA-based MIPs were designed by Gagliardi et. al. for the recognition and delivery of biotin. Their good biodegradability not only provides adjustable release kinetics and ensures complete drug release upon hydrolysis, but also lead to superior biocompatibility as a consequence of their lack of accumulation [6]. PEG is another synthetic macromolecule frequently used as a biocompatible hydrophilic scaffold for drug delivery. However, conflicting data about the impact PEG coating on the biocompatibility of the NPs is found in the literature. For example, one study showed that PEG1100-coated NPs induced mitochondrial toxicity, most likely due to its different conformation compared to PEG4000, which has proven to be very biocompatible [51].

Another topic that has recently been studied in connection to MIPs’ biocompatibility is the influence of PC. It has been recurrently shown that the in vivo fate and biological function of the nanocarrier relies on PC formation. The in vivo applications of MIPs should always consider PC formations [7]. According to a recent study, the formation and composition of PC is a key factor in the clearance of NPs from the organisms, rendering such studies highly recommended in the future [7]. The PC can coat and shield the nanocarrier; therefore, its release profile could be severely altered. PC could drastically reduce the burst effect that comes from desorption of surface-adsorbed drugs. However, some reports related to polymeric nanoparticles mentioned that PC only slightly altered their drug release profile [52]. The main factors that affect particles’ biocompatibility are schematically presented in Figure 4.

**Table 2 ijms-23-14071-t002:** Summarized characteristics of MIPs that have reached in vivo studies.

* **Drug Reservoirs** *											
**No**	**Template**	**Monomer** **Cross-Linker**	**Polymerization Method**	**In Vitro Drug** **Release (%)**	**Animal Model**	**Route**	**In Vivo Efficacy Parameters**	**Clinical** **Applications**	**Advantages**	**Type of MIP-DDS** **Observations**	**References**
1	Insulin	MAA ± HEAA MBAA	PP H_2_O UV 12 h RT	90% > 700 min (pH 7.4)	Diabetic MWR	oral	Blood glucose 60%↓ in 2–3 h	Diabetes mellitus	Lack of hypoglycaemic effect Great alternative to s.c. insulin Provides protection from protein degradation	Biomimetic system HEAA provides faster release	[27]
2	CAP	MAA + MPDE EGDMA	BP: 53° 4 h Toluene/ACN POSS	80% in 14 h	Healthy MWR	oral	Gastroretentive images BA = 168.9% Release > 12 h Tmax = 3 h	Colorectal, breast cancer	↑ floating properties ↑ gastric residence time No cytotoxicity on MCF-7	Floating LC-DDS Cooperative effect of MPDE + POSS	[29]
3	LVF	MAA + MPDE EGDMA	SIT: 53°, 48 h Chloroform 9-VA	~90% in 24 h	Healthy MWR	oral	BA = 578.9% Floating > 24 h	Respiratory, urinary, soft tissue, skin infections	↑ floating time ↓MWCNTs in vivo toxicity by ↑ clearance due to 9-VA functionalization	Floating LC-DDS Graft-to approach LC-MIP-MWCNTs	[32]
4	FB	4-VP EGDMA	In situ polymerization: 60°, 14 h Green solvents: [BMIM]BF4, ChCl/EG	~50% in 6 h	Healthy MWR	oral	BA = 143.3% Tmax = 4 h	Inflammatory disease, rheumatoid arthritis, rachitis, gout, osteoarthritis	Green chemistry approach with binary porogenic system No toxic effects	Doping SWCNTs into 3D structure of MIP	[33]
5	AG	MAA EGDMA	In situ polymerization: 65°, 24 h EtOH Fe^+3^-trimesic acid	~100% in 16 h	MWR	oral	BA = 143.3% Tmax = 5 h	Breast cancer	No cytotoxicity on MCF7 cells Release: slower, stable 12 h, less fluctuations, no sudden bursts	MIP- MWCNTs MOFs used for CNTs dispersion	[34]
6	S-PRNL	MAA EGDMA	Reactive pore-filling of bacterial cellulose membrane 3-MPS, DMF, 60°, 20 h	60% in 48 h	MWR	t.d.	Skin permeation studies Tmax = 8 h Cmax = 3x Release ↑ for S than R	Cardiovascular disease	↓toxicity and immunogenicity of chitosan release < 0.5 h testing Selectivity 4x higher for S Limited transport of R	Transdermal patch Chitosan gel reservoir MIP membrane for the enantioselective controlled release of S-enantiomer	[28]
7	GTX	NIPAm + AA MBAA	FP: 20 min, 90° DMSO	90% in 7–10 h	MWR	oral	Tmax = 1.5 h Plateau 2–10 h	FQ-sensitive infections	High BA Slow release at low pH (120 h) Fast release at 43°	pH/temperature sensitive (through NIPAm) Hydrogel-based MIPs	[35]
8	S-AML	MAA + MPDE EGDMA	53°, 24 h Toluene/isooctane	45% in 14–16 h	MWR + Nude mice	oral	Longer gastric residence time (>60 min) Plateau 1.5–22 h BA 188.5%	Cardiovascular disease	Fluorescence imaging in nude mice proved efficacy	Drug reservoir Floating LC-DDS Solvent-responsive	[36]
9	AMO NQA-Lpp20	AA MBAA	Inverse microemulsion polymerization H_2_O, hexane 2 h myristic acid	70% in 8 h	*H. pylori* bearing mice	oral	Fluorescence imaging intensity 0–30 min 30 min gastric residence time negative colonies and urease test	*H. pylori* infection	DDS penetrates under the mucus and avoids being flushed pH-independent ↑ drug loading burst effect for loading dose	Dual MIPs Diagnosis by bacteria culture, rapid urease test NQA modified with myristic acid for amphiphilicity	[23]
** *Targeted DDS* **											
**No**	**Template**	**Monomer** **Cross-linker**	**Polymerization method**	**In vitro drug** **release (%)**	**Animal model**	**Route**	**In vivo efficacy parameters**	**Clinical** **applications**	**Advantages**	**Observations**	**References**
10	DOX P32	NIPAm + TBAm + TFMAA MBAA	PP, SIT: RT 20 h H_2_O, TFE Silica core MIP shell	~40% (pH 6)	Nude mice 4T1 tumor models	i.v.	0.8x↓ tumour volume Antitumor effect identical to intratumor injection	Overexpressed P32 breast cancer	Only causes apoptosis in overexpressed P32 cells Targeted fluorescence imaging Good biocompatibility	Dual-imprinted theranostic tool MIP-coated fluorescent Fluorescent silica core-shell	[37]
11	DOX CD59	NIPAm+, TBAm+ AA MBAA	Free-radical, SIT: 24 h RT Stober’s sol-gel method, H_2_O Silica core MIP shell Ce6, GdQDs	27.4% in 72 h (pH 5.5)	bearing MCF-7 tumour model in BALB/c mice	i.v.	Inhibition of tumour growth Weight of tumour tissue Tumour volume >80% cell viability	Overexpressed CD59 tumours	Negligible toxicity to MCF-7 and LoVo cells ↑ biocompatibility Synergistic effect 19% > monotherapy	Dual-imprinted theranostic tool Chemo-/photodynamic synergy MIP-coated fluorescent silica core Paramagnetic GdQDs and photosensitizers (Ce6) act synergistically	[38]
12	DOX	Dopamine	SIT: 12 h RT Fe_3_O_4_ core PDE coating	90% in 8 h	Inbred BALB/C female mice		↓ tumour volume ↓ specific growth rate ↑ tumour doubling time ↑ tumour growth delay ↑survival time	Breast adenocarcinoma	↓ toxic effects ↑ DOX and Fe concentrations in tumour tissue, ↓in kidney and liver ↑ therapeutical effect and survival rate with external magnetic field Lack of non-specific toxicity	External magnetic guiding PDE coated Fe_3_O_4_ core Low toxicity Biocompatibility	[40]
13	OLZ	Fructose	Co-PP: 60° 20 h ACN/DMSO SiO_2_ coated Fe_3_O_4_ core	-	HR	i.v.	↑ brain concentration shows targeted delivery	Psychotic disorders	Fructose from degradation acts as energy source for brain cells Magnetic-guided drug delivery to the brain No trace of OLZ at 216 h	External magnetic guiding Biodegradable, biocompatible Low toxicity	[45]
14	5-FU	Tannic acid	Mini-emulsion polymerization: 70° 2 h ACN/hexadecane SiO_2_ coated Fe_3_O_4_ core	~70% in 120 h (pH 7.4)	HR	i.v.	Fluorescence images prove successful transport to liver	Lung, skin, colorectal, breast, brain liver cancer	MCF-7 cells multi anti-cancer performance Facile flow through vessels Magnetic-guided drug delivery to the liver	External magnetic guiding Biodegradable, biocompatible Non-toxic for live cells	[46]
15	5-FU	AA + MBAA EGDMA	PP: 24 h, 60° ACN/MeOH	~90% in 30 h	Female swiss albino mice bearing EAC	oral	↑ antitumor effect ↑ apoptosis ↓ tumour weight	Ehrlich ascites carcinoma	Significant down-regulation in tumoral expression caspase-3 and VEGF	PCR for quantitative gene expression of caspase-3 and VEGF	[53]
16	CAP	MAA + 4-VBBA TRIM	ACN 60°, 12 h	~80% in 10 h	WR	oral	Release > 11 h BA = 96.2%	Advanced metastatic breast, colon, rectal cancer, and solid tumours	No cytotoxicity of MCF7 cells	Association ↑ recognition ↑ BA due to boronic acid moiety	[39]

Legend: AA= acrylamide, ACN = acetonitrile, AMO = amoxicillin, AG = aminoglutethimide, BA = bioavailability, [BMIM]BF4 = 1-butyl-3-methylimidazolium tetrafluoroborate, BP = bulk polymerization, CAP = capecitabine, CD59 = CD59 epitope, Ce6 = chlorin e6, ChCl/EG = choline chloride/ethylene glycol, Cmax = maximum plasma concentration, CNTs = carbon nanotubes, Co-PP = co-precipitation polymerization, DDS = drug delivery system, DMF = dimethyl formamide, DMSO = dimethyl sulfoxide, DOX = doxorubicin, EGDMA = ethylene glycol dimethacrylate, EtOH = ethanol, FB = fenbufen, FP = frontal polymerization, FQ = fluoroquinolones, 5-FU = 5-fluorouracil, Gd = gadolinium, GTX = gatifloxacin, *H. pylori* = *Helicobacter pylori*, HEAA = N-hydroxyethyl acrylamide, HR = healthy rats, i.v. = intravenous, LC = liquid crystalline, LVF = levofloxacin, MAA = methacrylic acid, MBAA = *N*,*N*-methylene-bis(acrylamide, MeOH = methanol, MIP = molecularly imprinted polymer, MOFs = metal-organic frameworks, MPDE = 4-methylphenyl dicyclohexyl ethylene, 3-MPS = 3-methacryloxypropyl trimethoxysilane, MWCNTs = multi-walled carbon nanotubes, MWR = male Wistar rats, NIPAm = N-isopropyl acrylamide, NQA = N-terminal amino acid sequence 83–115, OLZ = olanzapine, P32 = P32 epitope, PCR = polymerase chain reaction, PDE = polydopamine, POSS = polyhedral oligomeric silsesquinoxanes, PP = precipitation polymerization, QDs = quantum dots, RT = room temperature, S-AML = S-amlodipine, S-PRNL = S-propranolol, s.c. = subcutaneous, SIT = surface imprinting technique, SWCNTs = single-walled carbon nanotubes, TBAm = N-tert-butyl acrylamide, t.d. = transdermal, TFMAA = trifluoromethacrylic acid, Tmax = corresponding time of Cmax, TRIM = trimethylolpropane trimethacrylate, 9-VA = 9-vinyl anthracene, 4-VBBA = 4-vinylbenzeneboronic acid, 4-VP = 4-vinyl pyridine, VEGF = vascular endothelial growth factor, WR = Wistar rats. The "↑", "↓"symbols represent an increase or decrease in the respective parameter.

## 6. Conclusions and Prospects for Further Development of MIPs as DDS

The current review aimed to describe the specific physicochemical properties that render MIPs suitable for clinical use, from both a material science and biopharmaceutical perspective.

In the light of the reported literature, MIPs have shown promising results, with high tolerability and therapeutical efficacy in various diseases; therefore, continuous efforts in further improving their multivalent features towards in vivo assessment and eventually routine clinical exploitation are worth considering. Rational design to achieve complex MIP-based drug delivery platforms brings significant contribution to the development of a more focused, efficient, and patient-centered therapy, getting one step closer to the concept of precision medicine.

Whether designed as drug reservoirs or targeted DDS, MIPs offer superior pharmacokinetics for various APIs, leading to higher bioavailability and superior safety profiles. Moreover, an MIP-based approach offers the opportunity to revisit certain APIs with limited use, or those that have been decommissioned in the past, due to poor bioavailability at the site of action, suboptimal efficacy, or tolerability.

The main concern regarding MIPs administration in living organisms is their biocompatibility. Current information on their safety and tolerability reveals several knowledge gaps and shortcomings in the conducted experimental designs, leaving some questions still to be addressed. Long-term biocompatibility and toxicology studies are still necessary to establish potential safety risks of MIPs under in vivo conditions and to make noteworthy steps towards their future medical use.

Despite the rather limited number of in vivo studies previously conducted on MIPs, the contributions made to the field of drug delivery discussed in this review are encouraging and should inspire the continuation of these efforts. Nevertheless, further advances in MIP design and synthesis should follow the “fit-for-purpose” concept, taking into consideration their potential clinical use from the very beginning of their development.

## Figures and Tables

**Figure 1 ijms-23-14071-f001:**
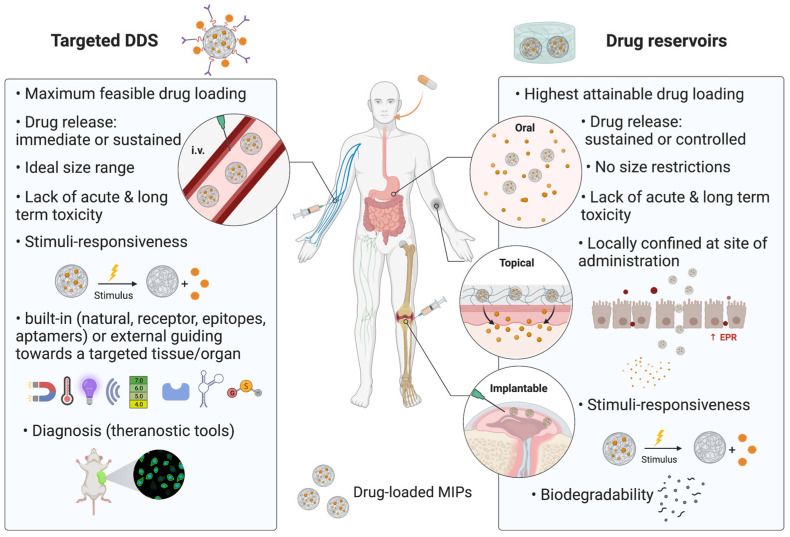
Comparison of the critical features of MIP-based drug delivery systems. The main differences and similarities between targeted DDS and drug reservoirs are shown in Figure 1. Both classes ought to lack toxic effects and can be designed as stimuli-responsive.

**Figure 2 ijms-23-14071-f002:**
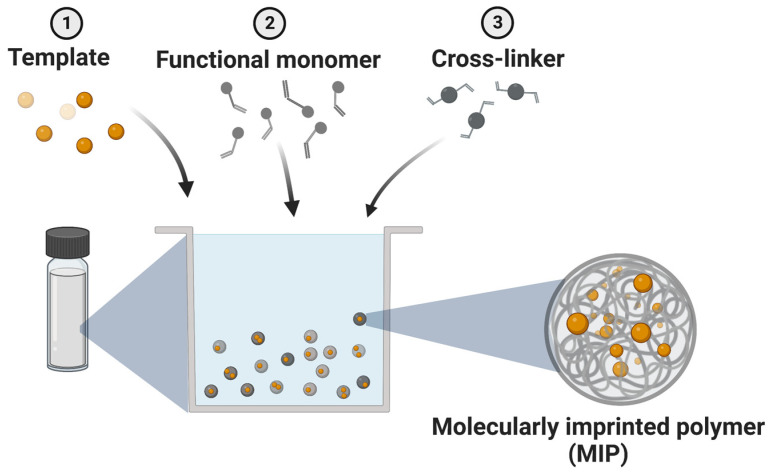
Schematic representation of MIP synthesis. The illustration presents the simplified process of molecular imprinting, wherein a mixture of template, functional monomer and cross-linker generates a highly cross-linked polymer, under certain initiating reaction conditions.

**Figure 3 ijms-23-14071-f003:**
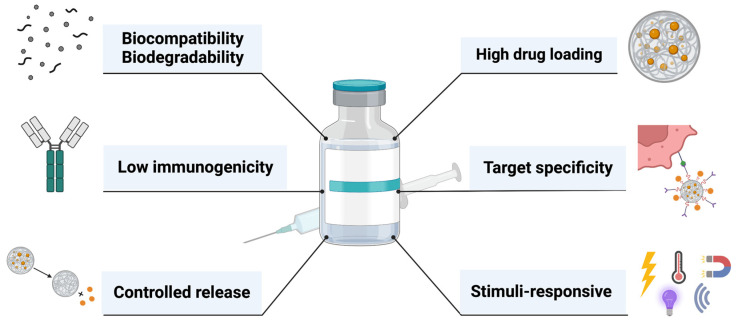
Features of an ideal MIP as DDS. This illustration shows the most important features for MIPs as DDS, which include their biocompatibility, biodegradability, low immunogenicity, controlled release, high drug loading, target specificity and stimuli-responsiveness.

**Figure 4 ijms-23-14071-f004:**
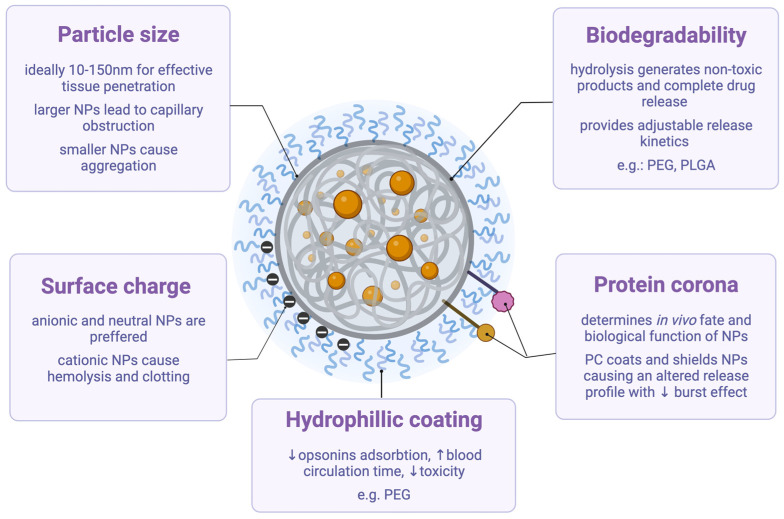
Factors affecting biocompatibility. This figure presents the main factors affecting the biocompatibility of nanoparticles (NPs), including their size, surface charge, hydrophilic coatings, potential biodegradability, and the formation of protein corona. PEG = poly (ethylene glycol), PLGA = poly-lactic glycolic-acid, PC = protein corona.

**Table 1 ijms-23-14071-t001:** Comparison of the characteristics of drug delivery systems based on fit-for-purpose concept.

Fit-For-Purpose	Drug Reservoir	Targeted Delivery
Drug loading	*Highest attainable*	*Maximum feasible*
Drug release	*Sustained/controlled release* simplified posology	*Immediate*, *sustained or stimuli responsive*
Size	*No size restrictions* Often confined at site of administration	*Ideal size range* EPR effect (enhanced permeability and retention effect); avoid various biological barriers and premature clearance in cancer treatment
Aim of functionalization	*Lack of acute and long-term toxicity*	*Built-in (natural receptor*, *epitope imprints*, *aptamers) or external (magnetic field*, *etc.) guiding towards a targeted tissue/organ*, Avoid first pass metabolization *Immune stealth feature* Avoid immune response *Stimuli-responsive drug release (photodynamic activation*, *local heating*, *pH change*, *biomarker*, *etc.)* *Lack of acute and long-term toxicity*
Route of administration	*Topical*, *implantable*, *oral*	*Intravenous*
Extra-features	*Stimuli-responsiveness*, *biodegradability*	*Diagnostics (theranostic tools)*, *biodegradability*

## Data Availability

Not applicable.

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
