# Peer review of "In Vivo Applications of Molecularly Imprinted Polymers for Drug Delivery: A Pharmaceutical Perspective"

_ijms, 2022, doi:10.3390/ijms232214071_

Round 1

Reviewer 1 Report

Baraian et al reviewed the molecularly imprinted polymers for drug delivery with potential in vivo application from a pharmaceutical perspective. 

Some aspects should be improved in consecutive paragraphs without any reference (e.g. line 233 to 262). 
Table 1 should be improved. It is not clear if this information is related to in vitro or in vivo studies. Table 1 should have references that support of information. 
The manuscript should be revised in terms of the numeration of tables (line 265). 

The authors should report more in vivo and clinical trial studies. Several in vitro studies are reported, however, studies should be compared and the differences and similarities should be highlighted.

Author Response

Reviewer #1:

Baraian et al reviewed the molecularly imprinted polymers for drug delivery with potential in vivo application from a pharmaceutical perspective. 

  1. Some aspects should be improved in consecutive paragraphs without any reference (e.g. line 233 to 262). 

The authors are thankful for the insightful observations made by the reviewers. Information from section 3 (lines 255 to 280, renumbered) represent the authors’ perspective on how MIP-based DDS should be classified. This consists of our personal contribution to the manuscript, so references are not imposed.

  1. Table 1 should be improved. It is not clear if this information is related to in vitro or in vivo 

Table 1 is meant to summarize and highlight the most important features of MIPs ought to be considered from their early design, as described in section 3, lines 255-280, irrelevant to their applicability in vivo or in vitro conditions. Table 1 synthesizes performance characteristics of MIPs based on the fit-for-purpose concept, as expected to be delivered in a real clinical setup.

  1. Table 1 should have references that support of information. 

The authors appreciate reviewer’s observation. Table 1 summarizes the authors’ perspective on how MIP-based DDS may be looked at and be classified. These represent original contributions (see above points 1. and 2.). Hence, references are not needed.

  1. The manuscript should be revised in terms of the numeration of tables (line 265). 

Thank you for the observation. Numeration was accurate in text, but the introductory paragraph for Table 2 was not placed accordingly. Table 2 can be found at line 660. The introductory paragraph for Table 2 has been repositioned after Table 1 to remove confusion on numeration (line 290). Table 2 introduces the reader for the following discussion on in vivo studies.

  1. The authors should report more in vivo and clinical trial studies. Severalin vitro studies are reported, however, studies should be compared and the differences and similarities should be highlighted.

The authors appreciate the recommendation. Within this manuscript, only in vivo studies were discussed (section 3, Table 2), as there are no reports on clinical trials on MIPs to this day.

In this review, the authors wanted to point out the limited number of in vivo studies and raise the awareness on the necessity to evaluate MIPs under in vivo conditions for further translation to clinical setup.

Reviewer 2 Report

This is an extensive review combining many different aspects of a cutting edge subject which are presented in an informative way. Below follow some suggestions for betterment.

1. The review could be significantly improved by the inclusion of more figures and the refinement of the existing ones. Some novel figures on what are molecularly imprinted polymers (MIPs) at the very beginning of the manuscript would help the non-initiated reader to get a quick grasp of the concept. The difference between DDS and drug reservoir is not that obvious. Please make clear with appropriate figure(s).

2. In Figure 1, please increase the size of the sketches representing "targeted DDS" and "tank reservoirs" (also for "drug-loaded MIPs"). We believe that a short explanatory text after the figure legend would add value to the figure. It could be even better if you divided this figure to more and included the new ones in the appropriate segments of the text.

3. Figure 2 is not self-explanatory. The message is not straight through and the presented sketches are too small. Please increase size of the sketches to be understandable and add a short figure legend to explain the content of the figure.

4. Have you considered making a figure for biocompatibility and biodegradability (see also our point 2)?

5. Language editing.

Line 60. “Biodegradable polymers have recently taken the spotlight in drug delivery”. Change to “Biodegradable polymers are recently under taken the spotlight in drug delivery” or “Biodegradable polymers are recently in the spotlight in drug delivery”.

Line 73. “Since often in MIP design the lack of interdisciplinary approach may be observed,” change to “As MIP design has been marked by a lack of an interdisciplinary approach,”.

Line 76. Delete entirely “to offer a critical review.”, end the sentence in “sciences.”

Line 82. Change “except there still are some challenges” to “except for some challenges”

Line 84 Change “proper selection of the template, solvent and functional monomer, which is guided by the…” to “proper selection of template, solvent and functional monomer, all guided by the…”

Lines 93-94. Change “Cross-linker’s importance is related to the preservation of the binding cavity’s shape fidelity,” to “The type of cross-linker is related to the preservation of the binding cavity’s shape,”.

Lines 102, 119, 132, 183 and elsewhere: remove hyphen (“-”).

Lines 105-106. “However, based on the aims of each application, synthesis should be optimized according to their distinctive particularities.”. This sentence is not understandable. Do you mean: “The type of each application requires however, a specific synthetic approach.”

Line 114. Change “While the imprinting factor is primordial for MIPs” to “While the imprinting factor is crucial for MIPs”

Line 132. Change “Because” to “Due”.

Line 138. Change “The main benefit while using SPS consists of low…” to “The main benefit of using SPS is their low…”.

Lines 181-183. Change “The first attempt at designing MIPs for drug delivery applications, reported in 1998, was based on imprinting theophylline with the use of methacrylic acid (MAA) as func-tional monomer and EGDMA as cross-linker.” to “The first designed MIPs for drug delivery applications, was based on imprinting theophylline with methacrylic acid (MAA) as the functional monomer and EGDMA as the cross-linker [insert reference].”

Lines 183-184. Change “Besides good recognition properties for the  ophylline vs. its structural analogues (caffeine), these MIPs were able to release theophyl line in a sustained way (5).” to “Besides the good recognition properties of  ophylline compared to  its structural analogue caffeine, these MIPs could  prolong the release of theophylline (5).”

Line 195. Change “MIPs have proven to be promising DDSs…” to “MIPs are promising DDSs..”

Line 198. Change “They have proven to be effective in various” to “They were effective in various”.

Line 346. Put the + sign after and not before the numbers.

Line 643. You write “The current review aimed to emphasize the critical characteristics that make MIPs…”. Do you mean “The current review aimed to the specific physicochemical properties/characteristics that make MIPs…”? Also in the abstract, line 22. “Critical” is not the right word, while only “characteristics” is a vague description.

Author Response

Reviewer #2:

This is an extensive review combining many different aspects of a cutting edge subject which are presented in an informative way. Below follow some suggestions for betterment.

  1. The review could be significantly improved by the inclusion of more figures and the refinement of the existing ones. Some novel figures on what are molecularly imprinted polymers (MIPs) at the very beginning of the manuscript would help the non-initiated reader to get a quick grasp of the concept. The difference between DDS and drug reservoir is not that obvious. Please make clear with appropriate figure(s).

The authors appreciate the suggestions. Requested figure on what are MIPs was added at the beginning (line 102, Figure 2) to increase visual impact, along with a short explanatory legend. More details regarding molecular imprinting are detailed in section 2.

The differences between drug reservoirs and targeted DDS are presented in Figure 1 (line 99) and described in section 1 (lines 49-60) and section 3 (lines 255 to 280). Moreover, the differences between these DDSs are nuanced in an additional paragraph (lines 50-55).

  1. In Figure 1, please increase the size of the sketches representing "targeted DDS" and "tank reservoirs" (also for "drug-loaded MIPs"). We believe that a short explanatory text after the figure legend would add value to the figure. It could be even better if you divided this figure to more and included the new ones in the appropriate segments of the text.

As requested, the sketches in Figure 1 (line 99) have been increased and the explanatory text can be found in below. More details can be found in section 1 (lines 50-60) and section 3 (lines 255 to 280). The authors have decided not to divide Figure 1, as it is intended to provide an overview on the two different MIP classes, comparing for their similarities and differences.

  1. Figure 2 is not self-explanatory. The message is not straight through and the presented sketches are too small. Please increase size of the sketches to be understandable and add a short figure legend to explain the content of the figure.

Thank you for the suggestions. The sketches in Figure 3 (renumbered, line 226) have been increased and a short explanation of the figure can be found below. Features of an ideal MIP as DDS are described in text at section 3. The authors consider that pictograms positioned beside the key terms are meant to explain the concepts in a simplified way, so supplementary explanations become redundant.

  1. Have you considered making a figure for biocompatibility and biodegradability (see also our point 2)?

The authors highly appreciate the useful recommendation. A new figure (with a short explanatory text) for biocompatibility has been added to the manuscript (Figure 4) based on the extensive discussion from section 5.

  1. Language editing.

The authors thank the reviewers for the recommendations. Language editing was done as requested. Changes can be observed by “Track changes” in Word Template.

Line 60. “Biodegradable polymers have recently taken the spotlight in drug delivery”. Change to “Biodegradable polymers are recently under taken the spotlight in drug delivery” or “Biodegradable polymers are recently in the spotlight in drug delivery”.

The sentence was edited as suggested (line 66).

Line 73. “Since often in MIP design the lack of interdisciplinary approach may be observed,” change to “As MIP design has been marked by a lack of an interdisciplinary approach,”.

The sentence was edited as suggested (line 79).

Line 76. Delete entirely “to offer a critical review.”, end the sentence in “sciences.”

The sentence was edited as suggested (line 82).

Line 82. Change “except there still are some challenges” to “except for some challenges”

The sentence was edited as suggested (line 86).

Line 84 Change “proper selection of the template, solvent and functional monomer, which is guided by the…” to “proper selection of template, solvent and functional monomer, all guided by the…”

The sentence was edited as suggested (line 87).

Lines 93-94. Change “Cross-linker’s importance is related to the preservation of the binding cavity’s shape fidelity,” to “The type of cross-linker is related to the preservation of the binding cavity’s shape,”.

The sentence was edited as suggested (line 109).

Lines 102, 119, 132, 183 and elsewhere: remove hyphen (“-”).

Hyphen was removed from line 144. However, the remaining hyphens are generated automatically from breaking words into syllables by Word MS.

Lines 105-106. “However, based on the aims of each application, synthesis should be optimized according to their distinctive particularities.”. This sentence is not understandable. Do you mean: “The type of each application requires however, a specific synthetic approach.”

The sentence was edited as suggested (line 121).

Line 114. Change “While the imprinting factor is primordial for MIPs” to “While the imprinting factor is crucial for MIPs”

The sentence was edited as suggested (line 130).

Line 132. Change “Because” to “Due”.

The sentence was edited as suggested (line 147).

Line 138. Change “The main benefit while using SPS consists of low…” to “The main benefit of using SPS is their low…”.

The sentence was edited as suggested (line 156).

Lines 181-183. Change “The first attempt at designing MIPs for drug delivery applications, reported in 1998, was based on imprinting theophylline with the use of methacrylic acid (MAA) as func-tional monomer and EGDMA as cross-linker.” to “The first designed MIPs for drug delivery applications, was based on imprinting theophylline with methacrylic acid (MAA) as the functional monomer and EGDMA as the cross-linker [insert reference].”

The sentence was edited as suggested (line 199) and reference was inserted (ref 21)

Lines 183-184. Change “Besides good recognition properties for the  ophylline vs. its structural analogues (caffeine), these MIPs were able to release theophyl line in a sustained way (5).” to “Besides the good recognition properties of  ophylline compared to  its structural analogue caffeine, these MIPs could  prolong the release of theophylline (5).”

The sentence was edited as suggested (line 202).

Line 195. Change “MIPs have proven to be promising DDSs…” to “MIPs are promising DDSs..”

The sentence was edited as suggested (line 215).

Line 198. Change “They have proven to be effective in various” to “They were effective in various”.

The sentence was edited as suggested (line 218).

Line 346. Put the + sign after and not before the numbers.

The sentence was edited as suggested (line 371).

Line 643. You write “The current review aimed to emphasize the critical characteristics that make MIPs…”. Do you mean “The current review aimed to the specific physicochemical properties/characteristics that make MIPs…”? Also in the abstract, line 22. “Critical” is not the right word, while only “characteristics” is a vague description.

The sentence was edited as suggested (line 675). At line 22, “critical characteristics” was changed to “specific features”, as requested.

Round 2

Reviewer 1 Report

No additional comments are necessary to add.